# How to Develop Drug Delivery System Based on Carbohydrate Nanoparticles Targeted to Brain Tumors

**DOI:** 10.3390/polym15112516

**Published:** 2023-05-30

**Authors:** Vladimir E. Silant’ev, Mikhail E. Shmelev, Andrei S. Belousov, Aleksandra A. Patlay, Roman A. Shatilov, Vladislav M. Farniev, Vadim V. Kumeiko

**Affiliations:** 1Institute of Life Sciences and Biomedicine, Far Eastern Federal University, 690922 Vladivostok, Russia; vladimir.silantyev@gmail.com (V.E.S.);; 2Laboratory of Electrochemical Processes, Institute of Chemistry, FEB RAS, 690022 Vladivostok, Russia; 3A.V. Zhirmunsky National Scientific Center of Marine Biology, FEB RAS, 690041 Vladivostok, Russia

**Keywords:** nanoparticles, natural polymers, polysaccharides, glioblastoma, blood–brain barrier, drug delivery

## Abstract

Brain tumors are the most difficult to treat, not only because of the variety of their forms and the small number of effective chemotherapeutic agents capable of suppressing tumor cells, but also limited by poor drug transport across the blood-brain barrier (BBB). Nanoparticles are promising drug delivery solutions promoted by the expansion of nanotechnology, emerging in the creation and practical use of materials in the range from 1 to 500 nm. Carbohydrate-based nanoparticles is a unique platform for active molecular transport and targeted drug delivery, providing biocompatibility, biodegradability, and a reduction in toxic side effects. However, the design and fabrication of biopolymer colloidal nanomaterials have been and remain highly challenging to date. Our review is devoted to the description of carbohydrate nanoparticle synthesis and modification, with a brief overview of the biological and promising clinical outcomes. We also expect this manuscript to highlight the great potential of carbohydrate nanocarriers for drug delivery and targeted treatment of gliomas of various grades and glioblastomas, as the most aggressive of brain tumors.

## 1. Introduction

Glioma is a type of brain cancer that is nearly incurable due to the ineffectiveness of chemotherapy drugs, invasive growth, and the blood–brain barrier (BBB) that limits drug delivery [1]. The standard treatment for glioblastoma is surgical resection, followed by simultaneous radiotherapy and chemotherapy with temozolomide (TMZ) for 6 weeks, followed by adjuvant therapy with TMZ for 6 months. In addition to TMZ, there are four FDA-approved chemotherapy drugs for glioblastoma and high-grade gliomas treatment (Figure 1): lomustine, carmustine, taxol (paclitaxel, PTX), and bevacizumab [2,3]. All these drugs are administered intravenously. In addition, carmustine is used in the form of Carmustine wafer implants. The wafer is based on polifeprosan 20 (a copolymer of 1,3-bis-(p-carboxyphenoxy) propane and sebacic acid), which is biodegradable and can be loaded with chemotherapy agents such as carmustine or TMZ for a limited topical application directly into the area where the tumor has been removed. This approach has shown a significantly higher therapeutic impact and a reduced number of side effects [4,5].

Drug delivery to brain tumors is complicated by multiple factors, including poor drug solubility, low drug concentration at the tumor site, irregular tumor vasculature, intratumoral hypoxia and selective permeability of BBB, difficulty of surgical access to the tumor focus, infiltrative tumor growth, and metastasis [6].

Recent advancements in nanotechnology have led to the development of nanoparticles (NPs) that can bypass the BBB, efficiently deliver drugs to the tumor site, and penetrate glioma cell membranes [7,8]. They should be less than 200 nanometers, or even less depending on the surface charge, chemical properties, structure [9], and nanomechanical properties [10]. 

Important advantages are NPs’ safety and biodegradability. Non-organic and metal NPs, such as gold ones, showed a significant toxic effect on cancer and healthy cells [11]. A critical property of drug delivery systems is their ability to instigate the reversible immobilization of active molecules and their controlled release, to provide a constant therapeutic concentration, to maximize efficiency and minimize negative side effects. Additionally, a good option to facilitate the non-invasive monitoring of glioma progression and treatment response is the presence of non-invasive imaging capabilities [12]. 

To transfer through the BBB, NPs need to stimulate receptor-mediated transcytosis, which can be activated by some glycoproteins and antibodies. There is evidence from experiments with liposomal or peptide carriers [13], and carbohydrate transporters [14] showing their efficacy in vitro and in vivo on xenograft models. Carbohydrate NPs have emerged as a promising tool for anticancer therapy due to their excellent biocompatibility, biodegradability, and ability to target specific cancer cells. NPs can be produced from various natural and synthetic polysaccharides such as cellulose and its derivatives [15], chitosan [16], pectin [17], hyaluronic acid [18], and alginate [19]. Carbohydrate NPs can be loaded with chemotherapeutic agents, peptides, and vectors to enable the target cytotoxic effect [20,21]. Several studies have shown that chemotherapy-loaded carbohydrate NPs can effectively inhibit tumor growth and metastasis in various cancer models. Moreover, polysaccharide NPs can also increase drug solubility and bioavailability, leading to better pharmacokinetics, reduced general toxicity even in drug-resistant tumors [22], and reduced the toxicity of metal NPs [23]. However, there is still no FDA-approved carbohydrate-NP-based therapy using their cytotoxic and cytostatic effects and drug delivery potential. Our review is devoted to emphasizing the potential of carbohydrate polymer NPs for anticancer therapy and drug delivery and discussing some new potential polymers for NP fabrication. We have provided a detailed analysis of the various polymer compositions for NP preparation and administration.

## 2. Carbohydrates as Carriers for Drug Delivery and Tissue Engineering

### 2.1. Glucose-Based

Cellulose is the most abundant polysaccharide in nature. It is water-insoluble and chemically stable, so one of the approaches for functional material design is a high-intensity mechanic or ultrasonic treatment and the production of cellulose nanocrystals (CNCs) characterized by various shapes and sizes [24]. Chemical stability supplemented with morphological durability and persistence leads to various applications of surface functionalization, changing the hydrophilicity and charge, and providing new interaction sites for bio-affinity and biodegradability [25]. CNCs are usually isolated from lignocellulose biomass by inorganic acid hydrolysis (Figure 1). Some papers describe the process with sulfur acid [26], hydrochloric acid [27], nitric acid in pure form, and combinations [28]. The shape and size of the CNC and a wide range of chemical modifications suggest many ways to use them for the treatment of cancer [29].

**Figure 1 polymers-15-02516-f001:**
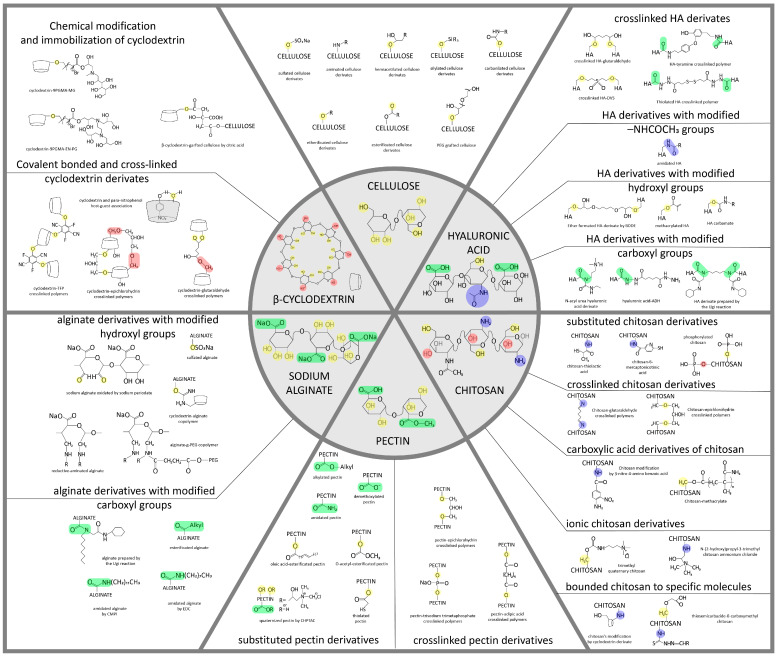
Structural formulas of some carbohydrates (cellulose and its derivatives, chitosan, pectin, hyaluronic acid, and alginate) and their possible modifications, on the basis of which it is possible to obtain various materials, including NPs [30,31,32,33,34,35,36,37,38]. The chemical modifications of polysaccharide nanoparticles are necessary to enhance their ability to cross the BBB and reach the target brain cells and for the attachment of targeting moieties and drugs. These modifications promote charge and active site setting, which allows nanoparticles with controlled drug release and biodegradation ability to be designed.

Cyclodextrins (CDs) are α-1,4-linked macrocyclic oligosaccharides obtained from cellulose by enzymatic hydrolysis. CDs are biocompatible, biodegradable, and non-toxic materials. The chemical structure of CDs is a truncated cone containing glucopyranose units with an outer hydrophilic exterior and inner hydrophobic pocket. It has been an attractive material for pharmaceutical applications due to this hydrophobic core to load various hydrophobic drug molecules by forming complexes through van der Waals force and hydrogen bonds. In that way, it is possible to overcome the undesirable physico-chemical properties of drugs: poor solubility and instability. Additionally, hydroxyl groups of the CDs cone’s exterior provide its solubility and the ability to form chemical modifications by functional molecules [39].

There are three most widely used CD native forms: α-CD, β-CD, and γ-CD, containing six, seven, and eight glucopyranose units, respectively. α-CD has a relatively small cavity that can only entrap small molecules. The drug capture of β-CD and γ-CD is better. Since γ-CD has a high production cost, β-CD with a moderate cavity and low production cost is the most widely applied CD in pharmaceutical research. The aqueous solubility of native α-CD, β-CD, and γ-CD is 13%, 2%, and 26% *w*/*w*, respectively. The low solubility of β-CD is due to the strong interaction of the inner hydrogen bonding formation among the secondary hydroxyl groups. This limit can be overcome by the disorganization of this strong hydrogen bonding [40,41]. 

### 2.2. Chitin and Chitosan

Chitin is the most abundant polysaccharide in nature after cellulose. It occurs in a number of eukaryotic species, such as crustaceans, mollusks, insects, and fungi. Chitin consists of 2-acetamido- (acetylated glucosamine, Figure 1 and Figure 2) and 2-amino-2-deoxy-β-D-glucoside (glucosamine, Figure 1 and Figure 2) residues connected by β-1,4-glycoside bonds. It is a homopolysaccharide, but usually, macromolecules include both types of residues [42].

Chitosan is derived from chitin by an N-deacetylation reaction. The reaction never goes through completely. The degree of deacetylation (DD) is always less than 95%. A polysaccharide with less than 50% deacetylated glucosamine residues is conventionally considered to be chitin, while a larger one is chitosan [43].

Chitin is soluble in a very limited number of organic solvents or under harsh conditions. The European Chitin Society suggests that a polysaccharide that is insoluble in 0.1 M acetic acid solution be considered chitin. Chitosan is soluble in aqueous solutions of most acids [44]. Its solubility is due to the polyelectrolyte effect because of charged amino groups in the macromolecule (Figure 1 and Figure 2) [45]. Polyelectrolytes are polymers containing functional groups capable of dissociation in aqueous solution. In this case, not one but many charges appear on their macromolecules.

Chitosan is known as a biologically active compound that exhibits numerous biological properties, such as antitumor, immune-strengthening, antifungal, antimicrobial, antioxidant, and wound-healing activities. The dependence of the effectiveness of the above properties on the DD was experimentally confirmed [46].

The above features as well as some exceptional properties such as non-toxicity, biodegradability, biocompatibility, and non-antigenicity have led to a wide application of this biopolymer in pharmaceutics, including biomedicine with the possibility of clinical use in drug delivery systems, tissue engineering [47], and food technology [48]. Moreover, interest in chitosan due to its biomedical applications in the central nervous system (CNS) has increased because of its potential ability to cross the BBB [49].

### 2.3. Pectin

Pectin primarily serves as a structural component of plant tissue’s cell wall. Citrus, apple, pear, and other fruits are used to extract pectin [50]. The chemical structure of this biopolymer has not yet been fully determined as it is a complicated natural molecule. Pectin is derived from linear polysaccharides, such as chains containing hundreds to thousands of saccharide units with average molecular weights ranging from 50 to 150 kDa. It is a heteropolysaccharide that consists of (1,4)-α-D-galacturonic acid residues branched with different neutral sugars (Figure 1 and Figure 3). There are four major building subunits, namely, homogalacturonan (~65% of pectin in plants), which consists of linear chains of α-1,4-linked methylated or acetylated galacturonic acid residues, rhamnogalacturonan-I (~20 to 35% of pectin in plants), xylogalacturonan (~10% of pectin in plants), and rhamnogalacturonan-II [51].

Pectin can be further classified by the ratio of esterified galacturonic acid groups to the total galacturonic acid group called degree of esterification (DE), which has a significant effect on the properties of pectin, particularly its solubility and gel formation, and biomedical and drug delivery applications [53].

Hydroxyl, amide, carboxyl, and methyl functional groups are the major functional groups in the pectin chain. Under different pH conditions, the -COOH of the pectin molecule can exist in different forms of -COO^−^, -COOH, and -COOH^2+^ [54]. Thus, pectin can shape into a composite with different feedstocks with opposite charges by adjusting the pH value of the medium.

Pectins that contain more than half of their carboxylate units as methyl esters have a relatively high carboxyl group ratio and are referred to as having a high methyl esterification rate. They are primarily used in gelation, require large amounts of sugar, and are very sensitive to acidity. Pectins that have less than 50% of the carboxylate units as methyl esters are usually referred to as having a low methyl esterification rate. This is obtained by slightly alkalizing a high ester pectin. It shows less sensitivity toward acidity and requires divalent ions to form gel [52]. Moreover, amidated pectin is produced through an alkaline process using ammonia to form a high ester polysaccharide. In this process, some of the remaining carboxylic acid groups are converted to acid amides. The properties of the amidated pectins can vary depending on the ratio of ester to amide unit ratio and the degree of polymerization. Polygalacturonic acid is partially esterified with methyl groups, and the free acid group is neutralized partially or completely with sodium, potassium, or ammonia ions [55].

Pectins have been reported to have a variety of bioactive properties including anticancer, anti-inflammatory, antioxidant, anti-diabetic, anti-cholesterol, anti-tumoral, and chemopreventive activities [54].

### 2.4. Hyaluronic Acid

Hyaluronic acid (HA) is a non-sulfated glycosaminoglycan consisting of disaccharides of D-glucuronic acid and N-acetyl-D-glucosamine, which are linked by β(1,4) and β(1,3) glycosidic bonds (Figure 1). HA is an anionic polymer (pKa = 3–4) [56], which allows it to interact with cationic polymers, surfactants, and lipids to form various nanostructures. HA has carboxyl and hydroxyl groups as well as an N-acetyl group, which gives a wide potential for further modification [57].

The key advantages of hyaluronic acid are its biocompatibility, complete biodegradability, and non-immunogenicity, which characterize hyaluronate as a biocompatible material for various biomedical applications. Interpenetrating networks of HAs facilitate the production of self-assembly aggregates, NPs, and hydrogels [58]. However, its short biological half-life and poor stability limit areas of its application [59]. In addition, HA fragments stimulate inflammation, tumor invasion, and P-glycoprotein-mediated multidrug resistance [60,61]. Fragments (25–50 disaccharide units) accumulate at the focus of inflammation and act as endogenous danger signals, reflecting oxidative stress in tissues [62]. HA oligosaccharides alter the expression of macrophages, chemokines, cytokines, and growth factors and can reduce the proliferation of endothelial fibroblasts and smooth muscle cells [63].

HA chemical modifications are mainly performed in aqueous solution involving two sites of the biopolymer: the hydroxyl and the carboxyl groups. Each disaccharide unit contains four hydroxyls, one amide, and one carboxyl group, and the resulting hydrogen bond associations prevent its hydrophobic modification in polar organic solvents. It limits the use of HA in the creation of permanent implants and other durable hydrophobic materials for biomedicine [64]. Alternative strategies are based on the partial protection of functional groups with hydrophobic blocking agents and the protection of anionic groups through complexation with cationic residues. Carboxyl groups are used for amidation and esterification, while hydroxyl groups form ester and ester bonds [65]. Dialdehyde groups formed after HA treatment with periodate can be used for chemical modification by reductive amination [66]. 

Hydrophobic modification allows for changing the rate of its degradation and hydrophilic properties depending on the degree of substitution. The attachment of amino acids with carboxyl or hydroxyl groups on HA provides strong resistance to its enzymatic cleavage. The biomedical application of HA is associated with its high hydrophilicity, unique rheological behavior, and its inherent pharmacological properties [67].

### 2.5. Alginate

Alginate is an anionic polysaccharide produced by brown algae and bacteria and consists of residues of α-L-guluronic acid (G-Block) and β-D-mannuronic acid (M-Block), linearly linked by 1,4-glycoside bonds. Alginate is non-toxic, biodegradable, low cost, and readily available, and has been found to be a mucoadhesive, biocompatible, and non-immunogenic substance. The composition and sequence of G and M residues depend on the source of the algae used and how this affects the properties of the alginate [68]. Alginate, a water-soluble unbranched polysaccharide, can also be chemically modified to change its properties (amidation, esterification, etc.) [32]. Alginate’s structure is shown in Figure 1 and Figure 4.

Alginates, which are anionic polymers with carboxyl groups, easily form gels in the presence of calcium ions and other divalent cations. Ionic and coordination crosslinking between alginate and divalent cations promotes interactions between alginate G-blocks and cations. Thus, alginates with a high content of guluronate can give more durable gels [70]. The addition of polycations such as chitosan or polylysine leads to the formation of a polyelectrolyte complex. At the same time, chitosan is considered preferable, since it causes a smaller immune response and is less toxic [71]. Alginates have an anti-anaphylaxis effect, immunomodulatory activity, antioxidant activity, and an anti-inflammatory effect [69]. The fields of application of alginate functional materials are regenerative engineering [72], wound dressing [73], and drug delivery [70,74]. 

## 3. Drug Delivery to the Brain

The standard of treatment for glioblastoma multiforme (GBM) is surgery followed by radiotherapy and chemotherapy. Despite these measures, the median overall survival of these patients ranges from 14.6 to 20.5 months [75]. Due to the invasiveness of this malignant neoplasm, local resection of the tumor only removes the primary tumor but cannot remove the surrounding tissues into which the tumor has already grown [76]. In addition, this treatment has significant drawbacks due to the systemic effects of chemotherapy drugs on the body. Therapeutic agents do not accumulate in the proper concentration in the tumor cells, thereby losing their effectiveness [77]. One of the major problems in therapy for GBM and other oncological brain diseases is getting drugs into the brain because the BBB prevents the transport of the most systemically delivered molecules [78]. Currently, there are a lot of investigations about targeted drug delivery methods into the brain. Some of them are about the targeted delivery of anticancer drugs by various types of matrix materials. This strategy of treatment, as well as radiosurgical boost after resection and the placement of a local radiation (brachytherapy) device in the resection bed following surgery, is the most prospective method of brain tumor treatment.

Strategies for drug delivery to the brain include invasive injection, temporary disruption of the BBB, and the use of drug delivery systems [79]. Temporary disruption of the BBB allows therapeutic agents to penetrate the brain. However, this approach can cause additional brain damage. Drug delivery systems are a less invasive and potentially safe method.

## 4. Invasive Strategies for Drug Delivery to the Brain

### 4.1. Intratumoral Drug Delivery

Intratumoral drug delivery for the treatment of glioblastoma involves local administration of therapeutic molecules by direct injection at the tumor site. One of the main directions of this method is the possibility of achieving a high concentration of drugs at the site of detection or resection while reducing the systemic exposure of drugs [80]. Direct injection is effective for local delivery in some cases, for example, with superficial tumor bedding, but it is absolutely ineffective in other cases, such as brain metastasis, when the distribution of therapeutic agents throughout the brain is required [81]. Local delivery of anticancer agents is effective in anti-glioma therapy due to the invasive nature of this tumor. Gliadel^®^ wafers represent a successful example of local intracranial drug delivery, resulting in improved survival of patients with newly diagnosed and recurrent glioblastoma multiforme [82,83].

### 4.2. Blood–Brain Barrier Transient Disruption

There are various methods for increasing transport through the BBB, such as destruction with osmotic shock, ultrasound, and magnetic gradient [79]. These techniques have several important limitations; this is often unfavorable for the patient and can compromise the integrity and physiological functions of the BBB, which can lead to the potential accumulation of unwanted blood components, neurotoxic substances, xenobiotics, and exogenous agents, leading to CNS damage [84,85].

Materials such as mannitol, fructose, and glycerol can induce high osmotic pressure by opening the BBB. The combined use of osmotic BBB disruption with chemotherapy for the treatment of malignant tumors of the CNS provides better progression-free survival compared to intravenous protocols with less hematological toxicity [86]. However, a change in the internal environment of the brain allows other macromolecular substances to penetrate the BBB, enter the brain, and cause neuropathological changes [85].

Convection-enhanced delivery (CED) is a promising method for localized drug delivery to brain tumors. This method has been proposed for delivering drugs that are either limited by the BBB or too large for efficient diffusion [87,88]. It is based on the positive pressure method and continuous infusion of the solution for a long time to maintain a constant pressure gradient and stimulate convection of the interstitial fluid in addition to the osmotic effect. CED requires only minimally invasive brain surgery [87,89,90] and can bypass the BBB by direct drug infusion into the interstitial space of the brain tumor through surgically placed catheters. Various types of therapeutic agents can be administered directly to a specific target area allowing large volumes of high-concentration drugs to be dispensed with minimal systemic toxicity. In addition, the use of this drug delivery method can lead to rapid coverage of large tumor volumes and a decrease in possible side effects [91,92]. Animal studies and clinical trials using this method have shown its effectiveness in the treatment of recurrent glioblastoma. However, the researchers note that longer and more in-depth studies and further optimization are required before it can be safely used in humans [93,94]. Even though the amount of distribution in the target tissue can be controlled, the geometric distribution in the desired target area varies greatly from one experiment to the next [95].

Ultrasound-assisted brain delivery is a minimally invasive method for targeted drug delivery to a brain tumor [96]. Microbubbles are pressed against the endothelial cell membrane and begin to vibrate under acoustic pressure. Vibration causes stress on the wall of endothelial cells, which leads to the temporary and local disruption of the BBB [97]. A combination of ultrasound-assisted brain delivery with traditional chemotherapy drugs, antibodies, NPs, and gene therapy significantly enhances brain uptake and therapeutic efficacy in many cases [98,99,100,101]. The therapeutic window for ultrasound-assisted brain delivery depends on the dynamics of BBB closure after rupture and the size of delivered molecules. The effectiveness of ultrasound therapy is highly dependent on tumor vascularization since blood vessels are key to the delivery of microbubbles and drugs to the tumor.

The disadvantage of implanted ultrasound devices is the need for invasive surgery. They are not aimed specifically at the tumor site and are mostly suitable for superficial brain tumors. It has also been found that in some cases, the ultrasound-mediated delivery of a specific monoclonal antibody increases brain uptake, but the therapeutic efficacy was not increased, which raised doubts about the practical use of ultrasound-mediated immunotherapy [102].

Magnetic drug targeting allows the therapeutic agent to be focused on a specific area of the body. In this method, chemotherapy drugs attached to magnetic NPs are injected into the blood vessels that feed the tumor and then are guided to the tumor site using an external magnetic field [103].

Invasive methods are difficult to use in diseases with a prolonged prodromal phase because they are also associated with relatively high maintenance costs and the requirement for follow-up, which contributes to patient noncompliance. They also have disadvantages such as poor penetration of the drug beyond the resection cavity, dosage of the drug limited by the size of the implant, connection with high intracranial pressure, and local toxicity causing infections and brain injuries [104]. Therefore, non-invasive delivery methods devoid of these disadvantages have been developed.

## 5. Noninvasive Strategies for Drug Delivery to the Brain

Non-invasive methods include viral vectors, NPs, exosomes, and delivery through active transporters in the BBB [105].

### 5.1. Viral Vectors

Viral vectors have a natural ability to infect cells with nucleic acids. The efficiency of viral vectors varies depending on the optimization of the transfection protocol, but in general, it is quite high. Vectors based on lentiviruses, the herpes simplex virus, adenoviruses, and adeno-associated viruses are used to deliver gene constructs to the brain. The practical use of viral vectors for drug delivery is limited because of the low safety of viral vectors due to patient deaths in clinical trials, production difficulties, and high production costs [105].

Several viruses have been developed as vectors for delivery to the CNS, including adeno-associated viruses (AAVs) (25 nm diameter), simian virus 40 (50 nm), lentiviruses (100 nm), and helper-dependent adenovirus (100 nm) [106]. Some AAV variants such as AAV-PHP.B and AAV-PHP.eB have been shown to facilitate a 40-fold increased delivery to the CNS following systemic administration in mice and to achieve a uniform distribution in the brain parenchyma, thereby providing a non-invasive alternative for gene delivery to the CNS [107]. Some variants also exhibit specific cellular tropism, which presents unique opportunities for effective drug targeting. For example, AAV-PHP.A for astrocytes and AAV-PHP.S for dorsal root ganglion neurons, cardiac, and enteric neurons [108].

Viral vectors, although effective, are associated with the risk of oncogenesis and mortality. In addition, viral capsid proteins and transgenes can spread to non-target tissues, which can lead to an exacerbation of the immune response, undesirable biochemical changes, and the transfer of vector DNA to germ cells [104].

### 5.2. Exosomes

Exosomes are a type of extracellular vesicle. Like all extracellular vesicles, exosomes play a role in intercellular communication through various receptors on their surface and have a specific set of components (proteins, mRNA, miRNA, and microRNA) that change depending on the state of the cell. Since exosomes are a natural element for the body and can be produced using cell lines, they have come to be considered as drug delivery vehicles to various organs, including the brain. Due to their properties, exosomes do not cause acute immunogenicity. They circulate in the blood for a long time and easily penetrate the BBB [109,110].

Salarpour et al. loaded PTX into exosomes isolated from the U-87 cell line and determined their cytotoxic effect in comparison with free PTX and empty exosomes. The size of the empty particles was 61 nm, loaded PTX—62 nm and 86 nm, depending on the method of loading. Loading efficiency turned out to be quite low: from 0.74% to 0.92%. Empty exosomes had little effect on cell viability (U-87 cell line): about 92% of living cells. Free PTX had a moderate effect: 80.7% of living cells. PTX in exosomes had a greater effect: 59.9% of living cells [111].

Erkan et al. used cytosine deaminase and uracil phosphoribosyltransferase in combination with 5-fluorocytosine as a suicide therapy for glioblastoma (in xenograft mice model). These enzymes convert 5-fluorocytosine to 5-fluorouracil, causing defective DNA replication and apoptosis. Previously, the authors showed the effectiveness of this approach for schwannoma tumors. This approach was also effective for glioblastoma. Tumor growth was reduced by 70% [112].

Although exosomes are a promising drug delivery method, there are several limitations. For example, there is no specific source of cells to produce exosomes, and therefore there is no standard protocol for isolation and purification. There are no specific protocols for loading various substances (proteins, DNA, RNA, etc.). It is not always clear how best to evaluate the efficiency of loading a substance [104,105,108].

### 5.3. Delivery through Active Transporters in the Blood–Brain Barrier

The BBB has many carrier molecules and their receptors through which substances enter the brain (glucose, amino acids, hormones, nucleosides, etc.). This method has several options: (1) active carriers (carrier-mediated transcytosis (CMT)) and (2) active carrier receptors (receptor-mediated transcytosis (RMT)) [85,104].

An example of the RMT strategy is the work of Hultqvist et al. [113]. The authors used a recombinant antibody containing two single-chain variable fragments of the 8D3 antibody to the transferrin receptor and mAb158 light chains. The last antibody can selectively bind to Aβ protofibrils that are involved in the pathogenesis of Alzheimer’s disease. This approach made it possible to transfer a sufficiently large molecule in the form of an antibody through the BBB, while the difference in the concentration of the antibody with and without a fragment to the transferrin receptor was 80 times. Thus, the authors showed that binding to various BBB active transport receptors increases its permeability to protein substances.

An example of a CMT strategy is the work of Lin et al. [114]. The authors used glutamate to target the large amino acid transporter or LAT1 to deliver docetaxel in liposomes. This transporter is overexpressed both on BBB cells and on glioma cells. Therefore, it was possible to create a dual-targeted drug delivery system: for penetration into the BBB and for targeting intracranial glioma. Compared to unmodified liposomes, there was a significantly higher cellular uptake and cellular cytotoxicity. Although the distribution in the brain region was less than in other organs, a significant difference was found in the brain region between the three groups: free dye DIR, DIR in liposomes with glutamate, and DIR in unmodified liposomes. No fluorescence was detected in the brain in the free DIR solution group, indicating that the solution cannot cross the BBB. However, fluorescence in the brain region was stronger in the groups treated with DIR-loaded liposomes than in the group treated with DIR solution, indicating that both modified liposomes and unmodified DIR-loaded liposomes may enhance accumulation. In addition, brain accumulation was much higher in the glutamate-containing liposome group than in the unmodified liposome group.

In the case of choosing an RMT for the transport of substances, bispecific antibodies targeting a specific receptor are used. The main problem is that target receptors can be found in many tissues, so it is difficult to choose such a substance so that it can effectively pass through the BBB and reach certain cells [115]. However, the implementation of the RMT strategy is more difficult than the CMT, because it is necessary to design antibodies with the least immunogenicity. The SMT strategy, in addition to being less specific, has a number of other disadvantages: for transporters with low to moderate transport capacity, the potential inhibition of drug uptake by high levels of endogenous plasma substrates must be considered; and resistance to enzymatic cleavage at the BBB interface observed in some nucleoside analogs may be an obstacle to achieving therapeutic levels of the drug in the brain [104,116,117].

### 5.4. Nanoparticles

NPs are colloidal systems in which the dispersed phase can be of natural or synthetic origin and can vary from 1 to 100 nm. According to some classifications, this also includes particles up to 500 nm and more [118]. There are many types of NPs, depending on the material and shape. Nanosized systems for drug delivery to the brain can be divided into lipid-based, polymer-based, and inorganic. Lipid-based NPs include liposomes, nanoemulsions, solid-lipid NPs, and nanocapsules. Polymer-based NPs include polymer NPs (synthetic and natural), micelles, nanogels, and dendrimers. Inorganic materials include quantum dots, metal, carbon, and silica NPs. The drug may be adsorbed on the surface of the particles, be within the particles, or be covalently bound to the particle [119]. The advantages of inorganic NPs include shape and size control, and the ease of preparation and functionalization. They are easier to track by various microscopy methods: magnetic resonance imaging, transmission electron microscopy, and others. However, not all of them can be degraded and eliminated. Most inorganic NPs can present a toxic effect. For example, carbon nanotubes and fullerenes can lead to lipid peroxidation and the formation of oxygen radicals. Natural NPs are produced from natural polymers such as polysaccharides including cellulose, chitosan, pectin, alginate (Figure 1), amino acids (poly(lysine), poly(aspartic acid), or proteins (gelatin, albumin). Natural NPs have the advantage of providing biological signals for interaction with specific receptors/transporters expressed by endothelial cells. Disadvantages are batch-to-batch variability, limited ability for controlled modification, and poor traceability with imaging platforms [118]. The main types of particles are shown in Figure 5.

## 6. Carbohydrate Nanoparticles for Brain Delivery

Carbohydrate NPs can deliver various small molecules such as PTX or venlafaxine [120,121]. In addition to small molecules, carbohydrate NPs can also deliver peptides [122,123,124] and siRNAs [125]. Recently, most works have been devoted to chitosan. Additionally, carbohydrates such as alginates, cyclodextrins, and cellulose have previously been considered [126]. Small molecules are more often used as therapeutic agents.

### 6.1. Cellulose-Based Nanoparticles for Brain Delivery

Cellulose NPs can be formed only after the pretreatment of the original polysaccharide by chemical, mechanical, or mechanochemical methods to obtain water-soluble analogs (Figure 1). In this way, either ready-made nanoscale spheres [127] or material for subsequent ionic gelation can be obtained.

Qian et al. made cellulose nanogels containing doxorubicin. The nanogels were tested on human neuroblastoma SH-SY5Y and mouse liver carcinoma H22 cell lines. The results showed high drug content and encapsulation efficiency. Doxorubicin was coupled to cellulose via disulfide bonds and these bonds were quite stable in serum for 96 h. Nanogels loaded with doxorubicin at a concentration of 16 μg/mL inhibited cell proliferation, while cell viability did not change. The doxorubicin gels showed greater inhibition of tumor growth and a higher degree of cellular apoptosis. Doxorubicin nanogels had longer circulation times and improved doxorubicin accumulation at the tumor site in vivo [128].

Dong et al. synthesized folic acid-conjugated cellulose nanocrystals for targeted delivery of chemotherapeutic agents to folic acid receptor-positive cancer cells. The authors chose human brain tumor cell lines (DBTRG-05MG, H4) and rat (C6) as model objects. Cellular binding and uptake of the conjugate by DBTRG-05MG, H4, and C6 cells was 1452, 975, and 46 times higher, respectively, than for non-targeted cellulose nanocrystals. The authors also established the mechanism of cellulose absorption by cells. DBTRG-05MG and C6 cells internalized the conjugate primarily through caveolae-mediated endocytosis, while H4 cells internalized it through clathrin-mediated endocytosis [129].

### 6.2. Cyclodextrin-Based Nanoparticles for Brain Delivery

CD has been used as a nanocarrier for drug delivery due to its ability to encapsulate hydrophobic compounds with high biocompatibility. However, it is difficult to deliver CDs into brain cells after crossing the BBB due to their hydrophilicity and relatively high molecular weight (ca. 1000 Da). There have been several sugar modifications of CD-nanocarriers for targeting GLUT1 to cross the BBB (Figure 1). Lactose-appended β-CDs showed great cellular uptake in hCMEC/D3 cells, high permeability within the in vitro human BBB model, and significant accumulation in the mouse brain [130].

Yokoyama et al. created various sugar-appended β-cyclodextrins (β-CyDs) to discover novel brain-targeting ligands. β-CyD supplemented with mannose, maltose, N-acetylglucosamine, galactose, glucose, and lactose were used to deliver complex β-CyD-HF647 into the mouse brain. TPPS (tetraphenyl porphyrin tetra sulfonic acid) was used as a model drug. HF647-Lac-β-CyD showed the highest accumulation in the brain during 1 h after administration. Then, it gradually decreased, and a slight accumulation was observed during 24 h after administration. HF647-Lac-β-CyD showed negligible accumulation in the kidneys, heart, lungs, and spleen. Lactose-appended β-CyD (Lac-β-CyD) showed greater cellular uptake in hCMEC/D3 cells and human brain microvascular endothelial cells than that of other sugar-appended β-CyDs. Moreover, Lac-β-CyD significantly accumulated in the mouse brain after intravenous administration. Thus, Lac-β-CyD efficiently facilitated the accumulation of the model drug in the mouse brain [130].

Gil et al. investigated β-cyclodextrin-poly-β-aminoether NPs to deliver doxorubicin to the brain showing the potential and the effectiveness of this formulation for brain drug delivery. NPs demonstrated sustained doxorubicin release for at least a month with a 100% BBB permeability coefficient without affecting the BBB model’s tight junctions’ integrity [131].

Cyclodextrins can also be used to deliver drugs to the brain bypassing the BBB to treat other non-cancer diseases. For example, to treat Alzheimer’s disease [132]. Crocetin is a potentially useful candidate for Alzheimer’s disease treatment but with poor water solubility and bioavailability. The crocetin-γ-cyclodextrin inclusion complex significantly increased the bioavailability of crocetin and facilitated CRT crossing the BBB to enter the brain [133].

### 6.3. Chitosan-Based Nanoparticles for Brain Delivery

Chitosan NPs can be obtained by different methods—emulsification with crosslinking, reverse micelle method, desolvation, reverse phase method, atomization, self-assembly of particles, ionic gelation, or ionotropic gelation [134]. Their advantages and disadvantages are listed in Table 1.

The latter two are mainly intended to be used for the treatment of tumor diseases. Using the ionic gelation method, positively charged amino groups of chitosan interact with negatively charged polyanions: sulfate ion and tripolyphosphate ion. Particle formation occurs as a result of cross-bonding of positively charged groups and negatively charged ions. Ionic gelation can occur simultaneously with radical polymerization, for example, of acrylic or methacrylic acids under the influence of potassium persulfate. The process involves the formation of NPs in conjunction with polyacrylic acid [135]. Assembly of chitosan macromolecules with oppositely charged polymer polyanions of natural or synthetic origin under the action of various interaction forces (electrostatic, hydrophobic, or van der Waals) to obtain polyelectrolyte complexes is possible (Figure 1) [136].

Chitosan polymers have shown a significant effect in antibacterial treatment [137]. Chitosan NPs have been used to deliver various drugs: selegiline hydrochloride [138], thymoquinone [139], and tramadol HCl [140]. Moreover, chitosan NPs were admitted for drug delivery for patients with Alzheimer’s disease, Parkinson’s disease, spinal cord injuries, cerebral ischemia, and nociceptive pain [126]. Chitosan NPs are also used to deliver drugs to brain tumor cells [141,142] (Table 2). Alipour et al. used chitosan NPs to deliver silibinin to C6 rat glioma cells to increase apoptosis. MTT analysis showed a greater number of cytotoxic effects for NPs loaded with silibinin than for free silibinin (approximately 15 times) and an increase in the expression of Bax and caspase3 genes, which are associated with apoptosis [143]. Chitosan NPs loaded with doxorubicin have been shown to target breast cancer cells and reduce tumor growth in vitro and in vivo [144].

Woensel et al. used chitosan NPs to encapsulate an anti-Gal-1 siRNA. Gal-1 is associated with tumor progression and is a potent immune suppressor in the tumor microenvironment. Loading siRNA into chitosan NPs saves it from degradation. The siRNA in the NPs was not degraded after incubation with RNases at 37 °C for several periods of time. In contrast, free siRNA rapidly degraded and could not be observed after 30 min of incubation with RNases. The delivered siRNA down-regulated the expression of Gal-1 in GBM and inhibited tumor progression in vivo [125]. NPs based on stearic acid and chitosan have shown promising results for intravenous delivery of DOX through the BBB [145].

### 6.4. Pectin-Based Nanoparticles for Brain Delivery

Pectin-based NPs have not been studied for targeted drug delivery to the brain through BBB. Nevertheless, the chemical structure of pectins and methods for their preparation are described in detail, thus making it possible to assess the prospects for their use. Pectin consists of at least three polysaccharide domains: homogalacturonan (HGA), rhamnogalacturonan-I (RG-I), and rhamnogalacturonan-II (RG-II). HGA is the main component of pectic polysaccharides and contains α-(1–4)-D-linked galacturonic acids (1,4-α-D-GalpA,) which are partially methyl-esterified or acetyl-esterified (Figure 1). The ratio of methyl-esterified residues (6-O-methyl-α-D-GalpA) of the HGA backbone to the total number of carboxylic acid units in salt form is called the degree of esterification (DE). The degree of esterification has been found to affect the properties of the gels formed and therefore must be carefully controlled to achieve the desired biomedical application [146].

The main method for obtaining pectin NPs is ion gelation. As in the case of chitosan, pectin is able to self-organize when gelling agents are added: polyvalent calcium, magnesium, manganese ions [147], or oppositely charged polyelectrolytes, for example, chitosan [148]. 

An important property of a molecule that ensures its passage through the BBB is its size. It can be adjusted by varying the conditions of pectin NPs preparation. The particle diameter depends on pH, presence of surfactants, nature of ions (Ca^2+^, Mn^2+^, and Mg^2+^), ion/pectin concentration ratio, and ionic strength [147,148]. Particle size decreased from 260 to 60 nm when the pH of the solution changed from 5 to 2.3. With the addition of 1% Tween-80, it decreased to 125 nm. The addition of NaCl had little effect. The authors [148] concluded that when using calcium and magnesium ions, in contrast to manganese, particles of the correct rounded shape are obtained. The particle diameter based on magnesium significantly decreased in comparison with calcium according to dynamic light scattering: ~600 nm and ~900 nm, respectively (Table 2). The authors also noted that the hydrodynamic diameter measured by the dynamic light scattering method is significantly larger than the size obtained after analyzing the transmission electron microscopy images. In the latter case, it was ~150 nm. The pH of the solution varied the zeta potential: at pH 2, it was −4 mV; and at 5.5, it was −54 mV.

Pectins have been evaluated as oral and nasal drug delivery particles. It was noted that the degree of pectin esterification affects the penetration depth of particles into the skin and mucous membranes, and it was also found that pectins can prevent the destruction of drugs loaded into them by the enzyme systems [149]. Several works have noted that pectin particles are a suitable base for drug systems against colon cancer [150,151]. This is because pectins have a higher contact time for drug action, which can also be useful in delivering drugs to the brain.

### 6.5. Hyaluronic Acid-Based Nanoparticles for Brain Delivery

HA is an essential component of connective, epithelial, and nervous tissue shaping the extracellular matrix and promoting cell migration and proliferation. It is an important material for NP-based delivery system design and its properties could be tuned by the covalent modifications (Figure 1). HA NPs can be produced using various methods. There are some common approaches.

1. Ionic gelation. In this method, hyaluronic acid and a cationic agent, such as magnesium or calcium ions [152], are mixed together in a solvent. This causes the formation of NPs due to the electrostatic interaction between the negatively charged hyaluronic acid and the positively charged cationic agent. The method is often used for the production of hyaluronic acid–chitosan NPs [153,154,155] or other combinations [156].

2. Coacervation. This technique involves the phase separation of hyaluronic acid from a solution. The NPs are formed due to the hydrophobic interactions between the hyaluronic acid molecules. This method requires the adjustment of pH, ionic strength, charge density, chain length, charge ratio, and may be used for HA–chitosan [157] or HA–protein NPs [158].

3. Emulsion/solvent evaporation. According to protocol [159], the drug and hyaluronic acid are dissolved in an organic solvent, which is then emulsified in an aqueous solution. The organic solvent is then evaporated resulting in the formation of hyaluronic acid NPs.

4. Reverse micelle. This method involves the formation of micelles of hyaluronic acid in an organic solvent, which then precipitates in an aqueous solution [160]. The choice of production method will depend on various factors such as the drug to be delivered, desired particle size, and functionalization of the NPs surface by the proteins or glycans [161].

HA NPs can help to reduce inflammation by binding with receptors on the cell surface such as CD44, toll-like receptors, and integrins that are involved in the inflammatory response via the mediation of the nuclear factor kappa-light-chain enhancer of activated B cells (NF-kBs), reducing the expression of inflammatory cytokines such as TNF-α and IL-1β, and increasing the production of anti-inflammatory cytokines such as IL-10 [162,163]. They have also been tested for their potential use in gene therapy, where they can deliver genetic material to target cells. There are some compositions based on combinations of hyaluronic acid with other polysaccharides including chitosan that might be used for targeted drug and gene delivery [164]. 

Hyaluronic and hyaluronic-embedded NPs have been investigated for delivering various drugs, including anticancer drugs, such as doxorubicin [165,166], PTX [167], and camptothecin [168]; anti-inflammatory agents, such as curcumin [169], quercetagetin [170], and dexamethasone [154]; antibiotics, such as chlorhexidine [171]; and growth factors, such as bone morphogenetic protein-2 [172] (Table 2). Hyaluronic acid is used as an active targeting ligand for brain tumors. For instance, hyaluronic acid-conjugated liposomes better target glioblastoma cells compared to other brain cells due to the higher expression of CD44 in tumor cells [173].

### 6.6. Alginate-Based Nanoparticles for Brain Delivery

Alginate NPs are widely used to encapsulate substances, protect them from the aggressive effects of digestive enzymes when administered orally, and also for packaging poorly soluble substances (Table 2). Thus, the nanoencapsulation of probucol significantly increases its concentration in blood plasma and brain compared to the drug without encapsulation [174]. In addition, due to their mucoadhesive properties, alginate-based NPs can be used for oral and intranasal transport of drugs to the brain. NPs loaded with therapeutic agents demonstrate complete biocompatibility and prolonged drug release [175,176].

For example, Haque et al. investigated the potential of venlafaxine-loaded alginate NPs (VLF AG-NPC) for the treatment of depression by intranasal delivery from the nose to the brain. Pharmacodynamic studies of the antidepressant activity of VLF AG-NPS were carried out in vivo using forced swimming and motor activity tests on albino rats of the Wistar line. Venlafaxine in NPs had a greater impact on behavioral analyses than venlafaxine in solution and in tablets. There was also a study of brain absorption. Pharmacokinetics was performed by determining the concentration of VLF in the blood and brain. Venlafaxine in NPs showed greater accumulation within 30 min in the brain than in other forms (free solution and tablets) [121].

Chavanpatil et al. used alginate particles ranging in size from 500 to 700 nm with the Aerosol OT^®^ surfactant to deliver doxorubicin to drug-resistant human breast cancer cells NCI-ADR/RES (MCF-7/ADR) and to drug-sensitive human breast cancer cells MCF-7, which overexpress P-glycoprotein (Pgp), which play the “gatekeeper” role in BBB.The authors used primary bovine brain microvessel endothelial cells (BBMECs) for a normal Pgp-overexpressing control. The authors showed the high rate of NP encapsulation on all Pgp-overexpressing cells including the normal BBB endothelial cells (BBMECs) without any significant effect of Pgp inhibitors [177]. We consider that this NPs formulation could be promising for high-dose drug administration to the brain from the blood vessels across the BBB endothelial cells.

The high biocompatibility and physicochemical properties of alginate-based NPs make it possible to use them for the delivery of CNS cells. Slowly degrading hydrogel provides protection for implanted cells from host immune responses while providing permeability to nutrients and released therapeutic molecules [178].

### 6.7. Polysaccharide Nanoparticles Effectiveness and Cytotoxicity

Chitosan NPs have shown significant therapeutic effects as drug delivery systems. There are a number of papers devoted to the development and implementation of chitosan-based NPs. Alipour et al. showed cytotoxic effects and induced apoptosis-related gene expression for silibinin-loaded chitosan NPs on a C6 rat glioblastoma cell line [143]. Kutlu et al. demonstrated an anti-tumor and anti-angiogenic effect in vitro [179]. Chitosan encapsulation may induce cytotoxic effects in doxorubicin-resistant gliomas [180]. TMZ-loaded chitosan/β-glycerophosphate thermogel microsystems showed a significant effect on inhibiting human glioblastoma tumor growth on orthotopic human xenograft models, which is expressed in 15-fold-reduced tumor growth in comparison to the control groups [181].

Xenograft models are commonly used for drug screening in brain tumor treatment due to their cost-efficacy and adequate modeling of the BBB. Schwinn et al. developed and tested a promising approach for medulloblastoma treatment based on axitinib/gemcitabine [182]. Osimertinib-loaded chitosan NPs demonstrated significant tumor growth inhibition compared to the control group (*p* < 0.01) in xenograft mice. The fabricated NPs demonstrated high stability and a possibility for their implementation for up to several weeks [183].

**Table 2 polymers-15-02516-t002:** Biological impact and properties of the common carbohydrate nanoparticle types.

	Nanoparticle Physical Properties				Nanoparticle Biological Impact		References
Polysaccharide	Size	Charge	Predominant Synthesis Method	Physical State	Drug Incapsulated	BBB Transmission Showed	
Chitosan	1–200 nm	Positive	Reverse micelle	Hydrogel	DOX, TMZ, lomustine, etc.	Yes	[141,142]
Pectin	25–900 nm	Negative	Ionic gelation	Hydrogel	TMZ, 5-Fluorouracil	No (for non-functionalized)	[184,185]
Alginate	10–150 nm	Negative	Ionic gelation	Hydrogel	TMZ, DOX, paclitaxel, etc.	No (for non-functionalized)	[186,187]
Cyclodextrin	Any	Uncharged	Ultrasonic emulsification	Hydrogel	5-fluorouracil, paclitaxel, TMZ	No (for non-functionalized)	[188]
Cellulose	Any, different-shaped	Uncharged	Ultrasonic treatment	Solid	DOX, tetracyclines	No (for non-functionalized)	[189,190]
Hyaluronic acid	Any	Negative	Ionic gelation	Hydrogel	Doxorubicin, PTX, camptothecin, curcumin, quercetagetin, dexamethasone, chlorhexidine	No (for non-functionalized)	[154,165,166,167,168,169,170,171]

However, the NPs delivery systems still require quality in vitro, ex vivo, and in vivo experiments to prove their efficacy. Only several studies have shown a significant effect in vivo, as well as significant BBB transmission (Table 2), so this approach still requires detailed quantitative and qualitative tests.

## 7. Conclusions and Future Perspectives

Carbohydrate polymer NPs have various potential applications in drug delivery and directly in glioma treatment, but the implementation of traditional non-modified carbohydrates for NP production limits its effect. NPs can be functionalized to enhance antitumor activity by chemical modifications changing their surface potential and affinity to cellular biopolymers.

One approach to using carbohydrate polymer NPs for glioma treatment is to encapsulate anticancer drugs within these NPs and target them to the tumor cells. These NPs can also be used to deliver the drugs across the BBB, which is often a challenge for drug delivery to the brain, but there is still a lack of data about barrier transfer efficiency and the possibility of the generalized effect on various organs and systems. To enhance the antitumor activity of carbohydrate polymer NPs, chemical modifications can be introduced to the surface. For instance, surface functionalization with targeting ligands such as antibodies or peptides can enhance the targeting specificity of the nanosized particles toward glioma cells. Furthermore, incorporating therapeutic molecules into the particle core can further enhance therapeutic efficiency. The important potential characteristics of NPs’ effectiveness are their therapeutic effects, their nanomechanical properties, and their involvement in mechanosensing. The significant role of NP nanomechanics on the effectiveness of drug delivery has recently been shown. Stiff materials display lower anticancer activity. However, there is still no significant evidence of BBB transfer effectiveness, which needs to be investigated in subsequent works. 

Novel studies should be focused on in vivo and ex vivo experiments, showing the effectiveness and cytotoxicity on cell models and xenograft models in addition to toxicity and biocompatibility assessments and bio-distribution profiling of functionalized carbohydrate-polymer NPs.

## Figures and Tables

**Figure 2 polymers-15-02516-f002:**
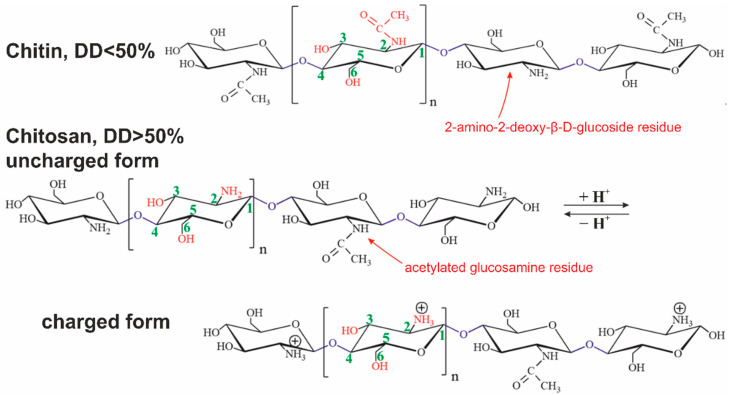
Structural formulas of chitin and chitosan in uncharged and charged forms. The presented polysaccharides can contain both types of glucosamine residues connected by O-glycoside bonds simultaneously.

**Figure 3 polymers-15-02516-f003:**
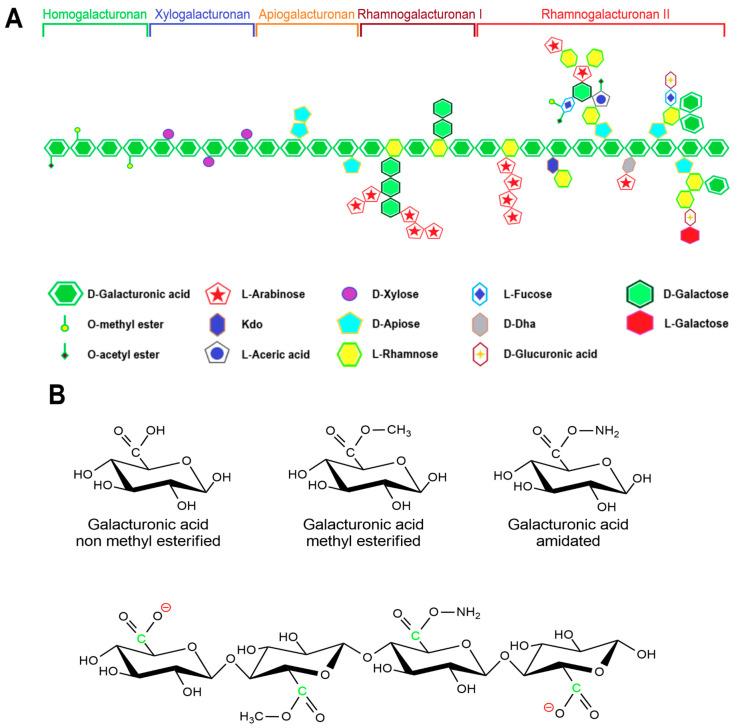
General scheme of pectins’ structure: (**A**) general image of pectins with structural units; (**B**) various modifications of galacturonic acid in the composition of pectins. The figure is based on data from [52].

**Figure 4 polymers-15-02516-f004:**
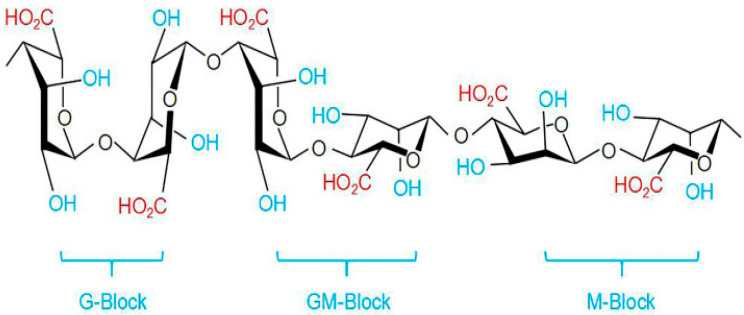
Structure of alginate based on data from [69].

**Figure 5 polymers-15-02516-f005:**
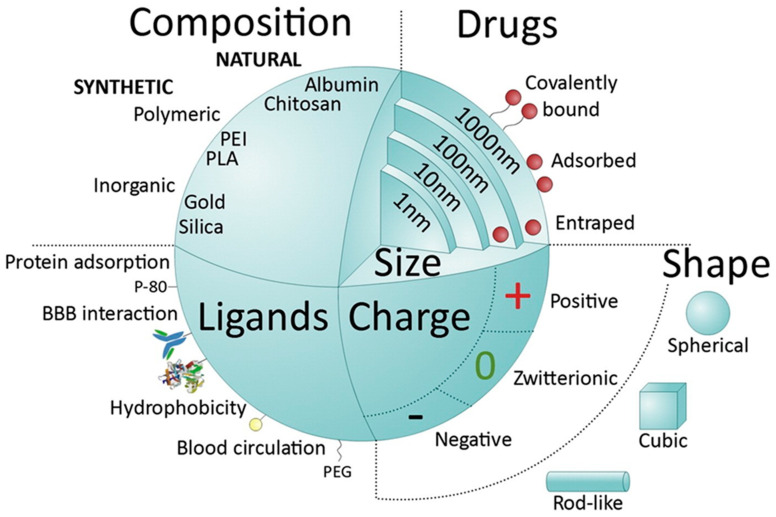
Main types of NPs and their parameters. Reprinted with permission from [118], copyright year 2023, copyright owner Elsevier, Journal of Controlled Release.

**Table 1 polymers-15-02516-t001:** General overview of chitosan NP preparation methods [134].

Method	Main Principle(s)	Advantage(s)	Drawback(s)
Emulsification and crosslinking	Covalent crosslinking	Simple process steps	Use of harmful chemicals
Reversed micelles	Covalent crosslinking	Ultrafine NPs below 100 nm	Time-consuming process Complex application Use of harmful chemicals
Phase inversion precipitation	Precipitation	High encapsulation capacity for specific compounds	Requires high shear force Use of harmful chemicals
Emulsion-droplet coalescence	Precipitation		Requires high shear force Use of harmful chemicals
Ionic gelation	Ionic crosslinking	Use of mild chemicals. Simple process. Ease of adjusting NP size	
Ionic gelation with radical polymerization	Polymerization and crosslinking		Time-consuming process Complex application
Self-assembly	Electrostatic and/or hydrophobic interaction	Highly stable NPs. Use of mild chemicals Adjustable procedure	Hard to control when carried out on a large scale
Top-down	Acid hydrolysis and deacetylation		Time-consuming process Complex application Need an extra step for drug loading
Spray drying	Atomization	Simple and fast process. Does not require another separation or drying steps	Large particle size Not suitable for use with temperature-sensitive substances
SCASA	Atomization	Acid- or harmful solvent-free method. Does not require another separation or drying steps	Time-consuming process Requires a specially designed system Large particle size

## Data Availability

Images and data are available from the corresponding author upon reasonable request.

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
