# Peer review of "How to Develop Drug Delivery System Based on Carbohydrate Nanoparticles Targeted to Brain Tumors"

_polymers, 2023, doi:10.3390/polym15112516_

Round 1

Reviewer 1 Report

Recommendation: Major revision

The manuscript reviewed recent outcomes in the area of NPs for treating brain tumors. The manuscript has been written well and concluded with decent outlooks. The authors have highlighted the area of development of carbohydrate nanoparticle based drug delivery. This is where the manuscript stands out with quality and will add to the existing reviews in the related area that cantered mostly on the structure-action characteristics of drugs. Therefore, I recommend accept with the following revisions.

Could the authors please add a sections separately on the: “in vivo, and ex vivo experiments, showing the effectiveness and cytotoxicity on cell models and xenograft models” which should be mostly based on recent research outcomes, otherwise the claim in the conclusion is quite vague  

Figure 1: It is not clear, plz change it to a better, clear and less informative one conveying only relevant/required messages

The weakest part is the writing, sometimes, the meaning the authors convey is quite far from what they actually want to say.

Line2: use of same verb twice in one sentence

Line 13 & 14: sentence is not meaningful

Line 18 & 19: Verb becomes adjective

Line 643: "effect" or effectiveness?/ efficacy?

Line 34: sudden appearance of Uppercase letters

Line 36: IUPAC nomenclature

Line 495: incomplete sentence

I found many recent papers are not cited which are based on in vitro cytotoxicity studies e.g.,

Schwinn, S., Mokhtari, Z., Thusek, S. et al. Cytotoxic effects and tolerability of gemcitabine and axitinib in a xenograft model for c-myc amplified medulloblastoma. Sci Rep 11, 14062 (2021). https://doi.org/10.1038/s41598-021-93586-x

 K, S.K., Choppala, A.d. Development and Optimization of Osimertinib-loaded Biodegradable Polymeric Nanoparticles Enhance In-vitro Cytotoxicity in Mutant EGFR NSCLC Cell Models and In-vivo Tumor Reduction in H1975 Xenograft Mice Models. AAPS PharmSciTech 23, 159 (2022). https://doi.org/10.1208/s12249-022-02314-9

The weakest part is the writing, sometimes, the meaning the authors convey is quite far from what they actually want to say.

Line2: use of same verb twice in one sentence

Line 13 & 14: sentence is not meaningful

Line 18 & 19: Verb becomes adjective

Line 643: "effect" or effectiveness?/ efficacy?

Line 34: sudden appearance of Uppercase letters

Author Response

1. Could the authors please add a sections separately on the: “in vivo, and ex vivo experiments, showing the effectiveness and cytotoxicity on cell models and xenograft models” which should be mostly based on recent research outcomes, otherwise the claim in the conclusion is quite vague  

Answer. Dear reviewer, thank you very much for your kind assessment of our work. We have revised our manuscript and added the section devoted to the nanoparticles effectiveness and cytotoxicity showed on cell models and xenograft animals (lines 837-858, Table 2).

2. Figure 1: It is not clear, plz change it to a better, clear and less informative one conveying only relevant/required messages

Answer. Figure 1 was simplified, and its legend was rewritten in accordance to make it more clear for understanding and discussed it better (lines 262-267). We consider that this figure is crucial for our manuscript due to it contains all polysaccharide nanoparticles highlights and its chemical modifications which is necessary to compare variable carbohydrate nanoparticle structures.

3. The weakest part is the writing, sometimes, the meaning the authors convey is quite far from what they actually want to say.

Answer. We have revised our manuscript to improve English style and spelling.

4. Line2: use of same verb twice in one sentence

Answer. We reviewed the abstract and corrected it (lines 10-22).

5. Line 13 & 14: sentence is not meaningful

Answer. We have amended the abstract to make it more readable and clear (lines 10-22).

6. Line 18 & 19: Verb becomes adjective

Answer. We have revised the sentence, checked and corrected the spelling (lines 18-19).

7. Line 643: "effect" or effectiveness?/ efficacy?

Answer. This mistake was made that distorted the meaning of this paragraph. The correct variant is  « Chavanpatil et al. used alginate particles ranging in size from 500 to 700 nm with the Aerosol OT® surfactant to deliver doxorubicin to drug-resistant human breast cancer cells NCI-ADR/RES (MCF-7/ADR) and to drug-sensitive human breast cancer cells MCF-7 which are overexpressing P-glycoprotein (Pgp) which play the “gatekeeper” role in BBB. As a normally Pgp overexpressing control authors used primary bovine brain microvessel endothelial cells (BBMECs). Authors shown the high rate of NPs encapsulation on all Pgp-overexpressing cells including the normal BBB endothelial cells (BBMECs) without any significant effect of Pgp inhibitors [174]. We consider that this NPs formulation could be promising for the high-dose drug administration to the brain from the blood vessels across the BBB endothelial cells.» (lines 639-648).

8. Line 34: sudden appearance of Uppercase letters

Answer. We have reviewed and improved the style (line 35).

9. Line 36: IUPAC nomenclature

Answer. We added extra explanation of the polifeprosan 20 structure, role and application in brain tumor treatment (lines 35-40).

10. Line 495: incomplete sentence

Answer. We have amended the paragraph to make it more clear (lines 489-493).

11. I found many recent papers are not cited which are based on in vitro cytotoxicity studies e.g.,

Answer. Schwinn, S., Mokhtari, Z., Thusek, S. et al. Cytotoxic effects and tolerability of gemcitabine and axitinib in a xenograft model for c-myc amplified medulloblastoma. Sci Rep 11, 14062 (2021). https://doi.org/10.1038/s41598-021-93586-x.

 K, S.K., Choppala, A.d. Development and Optimization of Osimertinib-loaded Biodegradable Polymeric Nanoparticles Enhance In-vitro Cytotoxicity in Mutant EGFR NSCLC Cell Models and In-vivo Tumor Reduction in H1975 Xenograft Mice Models. AAPS PharmSciTech 23, 159 (2022). https://doi.org/10.1208/s12249-022-02314-9.

We have added the section devoted to the nanoparticles effectiveness and cytotoxicity showed on cell models and xenograft animals. Additionally, added new material to reflect the novel directions in this sphere (lines 837-858).

The final manuscript is attached. We hope that we have fully answered your questions and made all necessary changes.

Reviewer 2 Report

This review article surveys the recent examples of carbohydrate-based (cellulose, cyclodexdrin, hyaluronic acid, chitosan, alginate, pectin, etc.) nanoparticles for BBB penetration toward glioma drug delivery. The review is comprehensive and timely. The references cited are balanced. It is recommended for publication with minor revision.

Comment:

It will be clearer for the readers to have a table summarizing all (or most important) nanoparticle examples with their physical properties (type, size, drug, drug loading%, etc.) and the pharmacological/biological properties (cell/animal type, method of delivery, efficacy, etc.).

Author Response

It will be clearer for the readers to have a table summarizing all (or most important) nanoparticle examples with their physical properties (type, size, drug, drug loading%, etc.) and the pharmacological/biological properties (cell/animal type, method of delivery, efficacy, etc.).

Answer. Dear reviewer, thank you very much for your kind assessment of our work. We have added a table summarizing the most important physical and pharmacological properties of nanoparticles (Table 2).

The final manuscript is attached. We hope that we have fully answered your question and made all necessary changes.

Reviewer 3 Report

The purpose of this paper is to introduce the great potential of carbohydrate nanoparticles derived from natural sources to deliver drugs for the target treatment of the nearly incurable type of tumors – gliomas, and to review the research, requirements, effectiveness and challenges of this technology in in the process of creating nanoscale delivery systems.

Here are the questions raised about the review and suggestions for revision:

1. The title and abstract of this paper emphasize that this paper introduces the role of carbohydrate nanoparticle drug delivery system in tumor therapy, but it reviews the research progress of the structure and function of different carbohydrates in a large amount of space. The point is not clear.

2. The references cited in the introduction of the nanoparticles are not enough to support the topic of the article, and are long in time, unable to reflect the latest research progress, and the content is not comprehensive enough.

3. Chapter VI describes the current research results of carbohydrate nanoparticles for brain delivery. However, in the subsequent narrative process, there is no comprehensive introduction of how to apply these findings to the actual tumor treatment.

4. This paper has detailly described both noninvasive and invasive strategies for drug delivery to the brain, but it does not mention the advantages of carbohydrate nanoparticles for brain deliver compared to the previous two.

5. There are some references for your modifications (CHEMICAL SOCIETY REVIEWS, DOI: 10.1039/c2cs35259a; Advanced fiber materials, DOI: 10.1007/s42765-022-00223-x; Biomaterials, DOI:10.1016/j.biomaterials.2023.122029)

Moderate editing of English language is necessary

Author Response

1. The title and abstract of this paper emphasize that this paper introduces the role of carbohydrate nanoparticle drug delivery system in tumor therapy, but it reviews the research progress of the structure and function of different carbohydrates in a large amount of space. The point is not clear.

Answer. Dear reviewer, thank you very much for your kind assessment of our work. We have revised our manuscript and added the section devoted to the nanoparticles effectiveness and cytotoxicity showed on cell models and xenograft animals (lines 837-858). Additionally, we have revised the abstract to emphasize the goals more clearly (lines 10-22).

2. The references cited in the introduction of the nanoparticles are not enough to support the topic of the article, and are long in time, unable to reflect the latest research progress, and the content is not comprehensive enough.

Answer. We have revised and shorten a bit the introduction chapter to emphasize the carbohydrate polymer nanoparticles application in drug delivery (lines 27-270).

3. Chapter VI describes the current research results of carbohydrate nanoparticles for brain delivery. However, in the subsequent narrative process, there is no comprehensive introduction of how to apply these findings to the actual tumor treatment.

Answer. We have revised the Chapter VI to support it with the discussion of the novel methods of the nanoparticle administration and application on in vitro and in vivo models (lines 837-858, Table 2).

4. This paper has detailly described both noninvasive and invasive strategies for drug delivery to the brain, but it does not mention the advantages of carbohydrate nanoparticles for brain deliver compared to the previous two.

Answer. Carbohydrate nanoparticles are the delivery systems which could be delivered both invasively (directly to the brain, for example, during the surgery) or non-invasively as a part of chemotherapy and combined radio/chemotherapy. We revised the manuscript in accordance to make the manuscript being more readable and clearer.

5. There are some references for your modifications (CHEMICAL SOCIETY REVIEWS, DOI: 10.1039/c2cs35259a; Advanced fiber materials, DOI: 10.1007/s42765-022-00223-x; Biomaterials, DOI:10.1016/j.biomaterials.2023.122029)

Answer. We studied the recommended articles and cited them in our work. Also, we added new material to reflect the novel directions in this sphere (lines 669-673, 826-835).

The final manuscript is in the attachment. Hope that we have fully answered your questions and corrected your remarks.

Round 2

Reviewer 1 Report

The revised manuscript has addressed all the concerns with efforts from the authors to improve other uncharted errors as well. Therefore, I suggest accept in present form.